# Progress on Material Design and Device Fabrication via Coupling Photothermal Effect with Thermoelectric Effect

**DOI:** 10.3390/ma17143524

**Published:** 2024-07-16

**Authors:** Shuang Liu, Bingchen Huo, Cun-Yue Guo

**Affiliations:** School of Chemical Sciences, University of Chinese Academy of Sciences, Beijing 100049, China; liushuang23@mails.ucas.ac.cn (S.L.); huobingchen17@mails.ucas.ac.cn (B.H.)

**Keywords:** photothermal effect, thermoelectric effect, photothermoelectric effect, polymer composites

## Abstract

Recovery and utilization of low-grade thermal energy is a topic of universal importance in today’s society. Photothermal conversion materials can convert light energy into heat energy, which can now be used in cancer treatment, seawater purification, etc., while thermoelectric materials can convert heat energy into electricity, which can now be used in flexible electronics, localized cooling, and sensors. Photothermoelectrics based on the photothermal effect and the Seebeck effect provide suitable solutions for the development of clean energy and energy harvesting. The aim of this paper is to provide an overview of recent developments in photothermal, thermoelectric, and, most importantly, photothermal–thermoelectric coupling materials. First, the research progress and applications of photothermal and thermoelectric materials are introduced, respectively. After that, the classification of different application areas of materials coupling photothermal effect with thermoelectric effect, such as sensors, thermoelectric batteries, wearable devices, and multi-effect devices, is reviewed. Meanwhile, the potential applications and challenges to be overcome for future development are presented, which are of great reference value in waste heat recovery as well as solar energy resource utilization and are of great significance for the sustainable development of society. Finally, the challenges of photothermoelectric materials as well as their future development are summarized.

## 1. Introduction

With the rapid development of economic society, the demand for fossil fuels has increased dramatically. In order to meet the fast-paced industrial growth, not only are large amounts of natural energy consumed, but also the greenhouse effect and environmental pollution are increased. Therefore, the development of green, clean, and especially renewable energy sources holds a pivotal position in the sustainable development of human society [1,2]. The collection and conversion of renewable energy sources in the environment, such as wind energy [3], water energy [4], solar energy [5], and bioenergy [6], into electricity is an important solution. Among them, solar power is a new type of clean energy that can convert solar energy into electricity [7,8]. Solar cells are widely used in people’s daily lives due to their convenient use, sufficient energy, and no need to worry about power supply [9]. Of significant note, Su et al. found that most of the solar cells with high photovoltaic conversion efficiency are lead-containing chalcogenides, which are prone to heavy metal pollution and do not comply with the standard of green energy due to lead leakage and the simultaneous release of CO_2_ during processing [10]. 

However, the emergence of the photothermoelectric effect, an energy conversion method, can effectively solve the above problems. Light can be converted into heat directly, heat into electricity, or solar energy and heat energy at the same time into thermal energy [11,12,13]. It not only makes full use of solar energy but also improves the utilization of waste heat from the surrounding environment. The photothermal power conversion process includes two processes: the photothermal process and the thermoelectric process. Photothermal conversion is a common solar energy collection method that can obtain the highest solar energy conversion efficiency and is widely used in desalination [14], heating domestic water [15], purifying water [16], and power generation [17]. Despite the significant progress of solar heat-absorbing materials, there are still many difficulties, such as expensive materials used, complicated fabrication processes, and so on [18,19]. 

Thermal energy is widely present in our daily lives, but most of the energy is wasted. Methods to collect and utilize it to generate the required electrical energy while at the same time being in line with the policy of sustainable development have become a concern for researchers. With the development of technology, the discovery of the pyroelectric effect and the thermoelectric (TE) effect provides us with the theoretical support to collect thermal energy in the environment. Compared to the pyroelectric effect, the thermoelectric effect has been applied in real life due to its ability to perform thermoelectric conversion more effectively and efficiently [20]. The thermoelectric properties of TE materials are evaluated by the dimensionless thermoelectric figure of merit (*ZT*), which is expressed as: $$ZT=\frac{S^{2}\sigma T}{\kappa }$$
where *S*, *σ*, *T*, and *κ* represent the Seebeck coefficient, electrical conductivity, absolute temperature, and thermal conductivity, respectively [21,22]. Therefore, if high thermoelectric properties are required, then the material is required to have a high Seebeck coefficient and electrical conductivity with low thermal conductivity. However, it is difficult to obtain materials with very superior TE properties because it is difficult to decouple the three parameters [23]. Moreover, due to the low thermal conductivity of organic polymer materials, power factor (PF = *S*^2^*σ*) rather than *ZT* is often used to evaluate thermoelectric properties [24]. Meanwhile, how thermoelectric materials can be converted for energy harvesting in the absence of ambient temperature gradients has become a major obstacle limiting their development [25]. 

In summary, the photothermoelectric effect, based on the photothermal effect and the Seebeck effect, enables photovoltaic power generation and the conversion of light energy into electricity in the absence of spatial temperature gradients, which makes photothermal electrical generators (PTEGs) a popular research topic nowadays [26]. The PTEGs have the advantages of being small in size, light in weight, noiseless, and requiring no external equipment, which overcome the drawbacks mentioned above. By utilizing solar energy and the thermal energy of the surrounding environment, the photothermal generators can provide electrical energy under conditions with and without light [27,28,29]. In recent years, various measures have been proposed to improve the performance of PTEGs, which can be categorized into heterogeneous asynchronous and homogeneous asynchronous conversions (Figure 1) [30]. The latter is more suitable for use in practical production because of its advantages of low cost and easy processing. Here, we have organized the recent research progress of photothermoelectric materials and their related devices to provide a reference for researchers in related fields.

## 2. Photothermal Conversion Materials

Solar energy devices are commonly categorized into three groups based on their conversion methods: electrical energy (photovoltaic technology), thermal energy (solar thermo-collectors), and a hybrid of both [31]. Based on the photovoltaic effect, photovoltaic cells, or solar cells, are able to produce electricity when exposed to sunlight. However, the development of this technology has been constrained by the high cost of production, the use of toxic substances, and the complexity of preparation methods. Solar collectors, which convert solar radiation into the internal energy of the transport medium [32], are an important component of solar energy systems, absorbing solar radiation and converting it into thermal energy, which is used for a variety of purposes, such as product drying [33] and seawater purification [34]. In addition, photothermal agents can be used for drug release [35], cancer treatment [36], etc. through photothermal conversion.

One of the important issues to be considered in solar energy systems is the storage of thermal energy. The basic ways of storage can be categorized into sensible heat storage and latent heat storage [37,38]. Due to the fact that latent heat storage has a larger storage capacity and narrower temperature variation compared to sensible heat storage technology, it is thus gaining more and more attention [39]. Through continuous research, it has been found that mixing nanomaterials with the working fluid can improve the device’s photothermal conversion performance. These nanomaterials can be categorized into metal materials, metal oxides, carbon nanomaterials, and so on. Each type of material has its own application characteristics, and these three types of materials will be summarized next.

### 2.1. Metallic Nanomaterials

Metallic nanoparticles, including gold, silver, and copper, have promising applications in direct absorption solar collectors (DASCs) due to their inherent unique localized surface plasmon resonance (LSPR) properties that contribute to more efficient light absorption and scattering [40]. The collective oscillation of free electrons present in these metal conduction bands in response to an electromagnetic field is called plasma, and since the metal nanoparticles are smaller than the wavelength of incident light, a localized surface plasma is produced [41]. The plasma absorbs light energy and then transfers the energy into the lattice structure of the metal through electron-phonon coupling, thus generating heat. 

A number of recent studies have been conducted to explore the effect of different metal nanoparticles on DASC performance, demonstrating their potential for improving solar collector efficiency. Mallah et al. [42] synthesized silver nanorods and nanoplates of different sizes with aspect ratios of 4–9 in the furnace in an attempt to obtain nanofluids absorbing solar radiation over a wide spectral range by investigating the optical properties of nanoscale silver with different morphologies. The photothermal conversion efficiency of the nanofluids was obtained by testing the performance of different silver nanomorphologies under simulated sunlight. They produced a hybrid nanofluid with a total addition of 0.94 ppm, and the efficiency of the DASC was more than 70% at a solar radiation concentration factor of ~2. Gong et al. [43] designed a type of three-layer cylindrical Ag–SiO–Ag composite nanoparticles. The composite nanoparticles can broaden the absorption band using both electrical and magnetic resonance modes, so they can excite three absorption peaks at 390 nm, 470 nm, and 780 nm, which improves the absorption performance of the nanoparticles. It was also found that the maximum absorption peak intensity (Q_abs_) of the nanoparticles rose from 4.90 to 11.16 with the addition of a silica layer, which is 2.28 times higher than that of silver nanoparticles. Gupta et al. [44] prepared a hybrid nanofluid based on gold nanoparticles in Azadirachta Indica leaf extract and verified the broadband absorption of the prepared hybrid nanofluid from the visible to the near infrared spectra by testing it under real outdoor conditions. This material was placed under sunlight for 2 h, and its maximum temperature gain was about 20 °C, reaching a maximum photothermal efficiency of nearly 62%, which is about 12% higher than that of water. Meanwhile, no signs of sedimentation or agglomeration were detected by exposure to repeated sunlight for several hours, indicating the high stability of the sample. Moreover, the natural extract is non-toxic, inexpensive, and easy to degrade, which is in line with the future development direction.

These studies demonstrate the immense potential of designing and synthesizing nanomaterials with specific properties to enhance solar energy absorption and conversion efficiency, providing a significant foundation for material development that focuses on nanotechnology in photothermal applications.

### 2.2. Metal Oxide Nanoparticles

In the field of solar energy utilization, enhancing photothermal conversion efficiency is a key technological challenge. To achieve this goal, a series of studies have demonstrated that the application of metal oxide nanomaterials and nanofluid technology can significantly improve the performance of solar collectors. Metal oxides such as copper oxide, aluminum oxide, and iron oxide can absorb solar radiation, but their efficiency is limited by factors such as band gap, band edge states, and band positions. The photothermal conversion efficiency of these metal oxides greatly depends on material composition, crystal structure, and surface properties [45,46]. 

Gupta et al. [47] attempted to study the thermoelectric properties of a direct DASC at full scale outside the laboratory. They added Al_2_O_3_ nanoparticles to water and made them flow in the form of a thin film over a glass absorber plate as a direct absorbing medium, investigating its effect on the efficiency of tilted direct absorption solar collectors under outdoor conditions. It was found that the improvement in efficiency was lower for Al_2_O_3_ nanoparticle concentrations above 0.005 vol%. Karami et al. [48] prepared copper oxide nanoparticles with different volume fractions using a mixture of water and ethylene glycol (70%:30% by volume) as the base fluid. The collector efficiency was tested at two different inner surfaces (black and reflective) of the base wall at volume flow rates of 54, 72, and 90 L/h (0.015−0.025 kg/s), respectively. At a flow rate of 90 L/h, the efficiency of the collector with a black inner surface was about 11.4% higher than that of the collector with a reflective inner surface using the base fluid. The collector efficiency was improved by increasing the nanofluid volume fraction and flow rate, with the nanofluid enhancing the collector efficiency by 9−17%. Balakin et al. [49] investigated an aqueous magnetic nanofluid for establishing photothermal convection in a laboratory-scale direct absorption solar collector. The nanofluid was obtained from 60 nm Fe_2_O_3_ particles dispersed in distilled water with concentrations ranging from 0.5 wt% to 2.0 wt%. The nanofluid containing up to 2.0 wt% of iron oxide nanoparticles achieved a velocity of 5 mm/s under a magnetic field of up to 28 mT, at which time the maximum thermal efficiency of the collector was equal to 65%. Considering the plasma absorption properties of Ag nanoparticles (NPs) and antimony-doped tin oxide (ATO) NPs in the visible and near-infrared spectra, Sreekumar et al. [50] synthesized ATO/Ag composite nanoparticles to achieve wider absorption in these regions and potentially improve the solar-to-thermal conversion (STC) efficiency. The maximum thermal efficiency of 63.5% and the maximum exergy efficiency of 5.6% were obtained by applying a direct absorption solar collector with optimized nanofluid at a flow rate of 0.022 kg/s. It was also found that both thermal and energy efficiencies increased with a rising flow rate. 

These studies demonstrate that utilizing specifically designed nanomaterials and nanofluid technology can effectively enhance the photothermal conversion efficiency of solar collectors, creating new possibilities for the advancement of solar energy technology.

### 2.3. Carbon-Based Nanomaterials

In this part, we examined the photothermal conversion efficiency of carbon-based nanomaterials, including carbon nanotubes, graphene, and graphite utilized in solar thermal collectors. These carbonaceous nanomaterials exhibit pronounced light absorption capabilities, attributed to their distinctive electronic structure and high aspect ratio. Compared to other nanomaterials, carbon-based nanomaterials present advantages such as lower density, a larger surface area, enhanced stability, and improved corrosion resistance. Moreover, the exceptional thermal conductivity of these materials results in nanofluids with superior thermal conductivity and heat transfer coefficients compared to conventional base fluids and nanofluids derived from different metals and metal oxides [51,52,53]. 

Ahmadi et al. [54] evaluated the thermal performance of a solar collector utilizing nanofluids based on graphene nanoplatelets (GNPs). The nanofluid samples were formulated by suspending GNPs at mass fractions of 0.01 and 0.02 wt% in water. The experimental findings revealed that the introduction of GNPs at 0.01 and 0.02 wt% loadings amplified the solar collector’s efficiency by 12.19% and 18.87%, respectively, relative to water, with an elevation in outlet fluid temperature to 340.5 K and 344 K. Such outcomes underscore the potential of GNP nanofluids to augment collector efficiency in a concentration-dependent manner. Zeng et al. [55] investigated binary nanofluids composed of SiO_2_/Ag plasmonic nanoparticles and multi-walled carbon nanotubes (MWCNTs) for solar heat transfer and conduction enhancement. Given the strong infrared absorption of MWCNTs and the visible spectrum absorption peaks of SiO_2_/Ag nanoparticles, the amalgamation of nanofluids with divergent spectral attributes resulted in hybrid nanofluids with augmented absorption across a broader solar spectrum. At equivalent nanoparticle concentrations, the binary nanofluids exhibited greater absorption than singular suspensions of MWCNTs or SiO_2_/Ag nanofluids. Said et al. [56] employed the surfactant sodium dodecyl sulfate (SDS) to disperse single-walled carbon nanotubes (SWCNTs), producing stable SWCNT/water nanofluids for the analysis of solar collector thermal efficiency. The study ascertained that elevating the inlet fluid temperature and adopting reduced flow rates significantly improved overall thermal performance. The experimental data indicated that incorporating 0.3 vol% SWCNT nanofluid bolstered the flat plate solar collector’s (FPSC) energy efficiency by 95.12%, achieving a peak efficiency of 26.25% in contrast to water’s 8.77%.

These research findings conclusively demonstrate that the integration of carbon-based nanomaterials, such as carbon nanotubes and graphene, in solar collectors notably enhances thermal efficiency, paving the way for advancements in solar thermal conversion technology.

## 3. Thermoelectric Materials

Thermoelectric materials are capable of interconverting between thermal and electrical energy and can be used to collect or recover solar thermal energy and low-grade thermal energy due to their inherent characteristics of no moving parts, long life, and no energy consumption [24]. Thermoelectric generators (TEGs) can convert temperature differences into voltage through the Seebeck effect or produce heat energy upon energization through the Paltier effect [57]. Based on their working principle, thermoelectric materials can be categorized into electronic thermoelectrics (e-TEs), ionic thermoelectrics (i-TEs), and thermo-electrochemical cells (TECs) [58], which will be described next (Figure 2).

### 3.1. Electronic Thermoelectric Materials

The Seebeck effect, which refers to the diffusion of electrons and holes resulting in the generation of a voltage across a material due to the presence of a temperature gradient, is manifested in all types of materials, of which the e-TEs are a representative class [59]. The n-type (electron-conducting) or p-type (hole-conducting) materials are defined according to the conducting particles. The energy conversion mechanism of electronic thermoelectric materials is shown in Figure 2a,b. Traditional inorganic thermoelectric materials, including Bi_2_Te_3_ [60] and SeTe_2_ [61], have been widely studied. For Bi_2_Te_3_ material, the addition of Sb and Se elements can optimize the carrier concentration and reduce the thermal conductivity, and it was found that Bi_2_Te_3_ doped with selenium is an n-type material while Bi_2_Te_3_ doped with antimony is a p-type material. Meanwhile, the optimal stoichiometry of p-type material is (Bi_1−x_Sb_x_)_2_Te_3_ (x = 0.75), and the optimal stoichiometry of n-type material is Bi_2_(Te_1−y_Se_y_)_3_ (y = ~0.15) [62]. However, the application of these elements is limited due to their toxicity. It has been found that elements with lower atomic numbers also have good thermoelectric properties and are non-toxic, environmentally friendly, and efficient, including different materials (such as sulfides [63], oxides [64], chalcogenides [65], graphene-based [66], silicon [67], polymers or composites [68]) or different structures (such as nanostructured materials [69], layered materials [70], 2D materials [71], and clathrates [72]), which have been extensively studied. 

Organic materials have great potential in the field of thermoelectricity due to their advantages of low cost, light weight, flexibility, and easy processing, so more researchers are focusing on organic polymer materials. Among recently reported thermoelectric materials, organic and organic/inorganic composites are gaining more attention due to their above-mentioned merits [73,74,75]. By now, the main organic components in TE composites include thiophene-based organic small molecules (OSMs), poly(3,4-ethylenedioxythiophene) (PEDOT), polypyrrole (PPy), polyaniline (PANI), and so on. The inorganic components are mainly carbon-based nanomaterials [76,77,78], such as carbon nanotubes, graphene nanosheets, etc. 

Fan et al. [79] prepared high-performance SWCNTs and polypyrrole (PPy) thermoelectric composites. Figure 3a–d,a’–d’ illustrates the preparation of PPy as well as acid-doped SWCNTs by three-phase interfacial electropolymerization and the obtaining of the two to-be-mixed solutions, respectively, after which the two were mixed to prepare PPy/a-SWCNT composite films. It was found that the chemical doping of SWCNTs significantly affects the thermoelectric properties of the composites, and the power factor (PF) value of the composites increases with an increase in SWCNT content. The maximum power factor of PPy/a-SWCNT composites was as high as 240.3 ± 5.0 μW m^−1^ K^−2^ at room temperature, and the electrical conductivity was 5707.5 ± 496.0 S cm^−1^, where the PF value may be the highest value for PPy and its composites at that time. The feasibility of preparing high-performance polymer thermoelectric composites by a dynamic three-phase interfacial electropolymerization method of chemically doped SWCNTs was demonstrated experimentally. Considering the advantages of OSMs, such as their well-defined structure, high purity, simple synthesis method, and low cost, they have a wide scope and great development potential in the field of thermoelectrics. Li et al. [80] synthesized benzothieno [3,2-b] benzofuran (BTBF) and its derivatives, BTBF-Br and BTBF-2Br, and prepared their TE composites with SWCNTs. It was found that the highest molecular orbital energy levels and band gaps (E_g_) of BTBF, BTBF-Br, and BTBF-2Br decreased gradually with the introduction of Br groups on BTBF. These changes significantly improved the Seebeck coefficient and power factor of OSM/SWCNT composites. Among them, the SWCNT composite containing 50 wt% BTBF-2Br has the smallest E_g_ of 4.192 eV and the Seebeck coefficient of 56.55 ± 0.58 μV K^−1^ and a power factor of 169.70 ± 3.46 μW m^−1^ K^−2^ at room temperature, which has the best thermoelectric properties and good flexibility, and the PF of the film after 100 bending cycles still maintains 70% of the initial value.

### 3.2. Ionic Thermoelectric Materials

Ionic thermoelectric materials (i-TEs) have anionic and cationic conductive media and are based on two principles: the thermal current effect and the thermal diffusion effect, the latter of which is also known as the Soret effect. The thermal current effect refers to the redox reaction between two electrodes at a temperature difference, while the Soret effect refers to the aggregation of ions at the ends of a sample by thermal diffusion in an ionic conductor or electrolyte, which produces an ion concentration gradient and generates a thermal voltage, usually with a high Seebeck coefficient (Figure 2d) [59]. Based on the characteristics of i-TEs themselves, they are usually more suitable than e-TEs for applications in low-temperature gradient scenarios. Due to the poor performance of single-component materials (e.g., electrolytes such as acids, bases, and salts and dissociated polymers such as poly(styrene sulfonic acid) (PSSH)), i-TEs are often made high-performance and flexible through the use of composites that utilize synergistic interactions of different components [81,82,83]. According to the common types of composites, they can be categorized into polymer-based i-TEs and small molecule-based i-TEs.

Zhao et al. [84] prepared and tested the Seebeck coefficient of a poly(ethylene oxide) (PEO)-NaOH gel electrolyte consisting of NaOH dissolved in liquid PEO at 11.1 mV K^−1^, which is more than 50 times higher than that of Bi_2_Te_3,_ along with a 2500-fold increase in energy conversion efficiency. Replacing water with organic solvents can be an effective way to increase the ionic conductor’s thermal voltage by enabling anions and cations to have different mobilities or concentrations. Ionic thermoelectric supercapacitors (ITESCs) are suitable for intermittent heat sources such as the sun because they are charged under a temperature gradient, and the stored electrical energy after removing the temperature gradient can be delivered to external circuits. Ionic gel ionic liquids (ILs) and polymer-based ionic conductor materials have much lower conductivity than electronic conductors, and He et al. [85] made ionic gels into a quasi-solid state by utilizing ILs and silica nanoparticles to overcome their low thermoelectric conversion efficiency. They can have high thermal voltage and ionic conductivity. At room temperature, the ionic Seebeck coefficient is 14.8 mV K^−1^, the ionic conductivity is 4.75 × 10^−2^ S cm^−1^, the thermal conductivity is 0.21 W m^−1^ K^−1^, and the ionic power factor is 1040.4 μW m^−1^ K^−2^. The optimum ionic ZT_i_ value can be as high as 1.47, which is almost twice the highest ZT_i_ value of ionic conductors previously used for thermoelectric conversion. 

### 3.3. Thermo-Electrochemical Cells

Thermo-electrochemical cells (TECs), also known as thermocells, utilize redox pairs to supply power to the outside. When there is a temperature difference between the two electrodes, the anions and cations at the electrodes will undergo the gain and loss of electrons, thus continuously supplying power to the outside, which is also the greatest difference from i-TEs (Figure 2c). Using carbon nanomaterials as electrodes, water or non-aqueous electrolytes with various redox pairs can give TECs advantages such as high surface area, fast electron flow, low cost, and easy processing [86]. However, the thermoelectric conversion efficiency of TECs still has a lot of potential for improvement, so it needs further study.

Kim et al. [87] proposed the use of a “solid-state” ionic conductor to maintain the temperature gradient and high ionic charge, i.e., PSSH was used as a solid electrolyte and heat collector, and two graphene and carbon nanotube films deposited with PANI (P-G/CNT) were used to sandwich the PSSH between the electrodes for storing electrochemical energy. This approach allows for the simultaneous generation of large voltages from temperature gradients and the storage of electrical energy while retaining the advantages of a device with no moving parts. PSSH films can be used to increase the output voltage (8 mV K^−1^) and conductivity (9 S m^−1^) by thermal diffusion, in addition to their measured ZT value of 0.4. At a small temperature difference of ΔT = 5 K, the thermally chargeable supercapacitor (TCSC) generates a potential of 38 mV and a large-area capacitance of 1200 F m^−2^, making it suitable for continuous power generation in wearable device electronics. 

Lu et al. [88] enhanced the PAA hydrogel electrolyte by introducing Mxene to enable multiple self-repairing and self-powering processes, and the fast gel properties, high flexibility, and adhesion of this electrolyte enabled the rapid assembly of flexible TEC arrays, thus avoiding the complex fabrication of typical wearable electronic devices. The resulting hydrogel-based TEC produces a maximum power output of 1032.1 nW at a ΔT of 20 K, which corresponds to 80% of its initial state when stretched to 500% for 1000 cycles, at which point the Seebeck coefficient was 2.33 mV K^−1^; meanwhile, it maintains a maximum power output of 1179.1 nW at a ΔT of 20 K, which is about 92% of its initial state, even after 60 cut-healing cycles. The assembled TEC array can be applied to touch-based encrypted communication, self-powered sensors, and so on.

Burmistrov et al. [89] investigated the efficiency of TECs based on nickel hollow microsphere electrode materials, which were pyrolyzed by ultrasonic spray pyrolysis and then reduced with the help of a hydrogen environment. Hollow Ni microspheres with a unique surface area and low electrical resistance (due to the high content of metallic phases) were investigated and considered as promising electrode materials for TECs. The results show that at 85 °C, this TEC provides a Seebeck coefficient of up to 4.5 mV K^−1^ and an open-circuit voltage of 0.2 V, which is suitable for commercial power supply circuits for various microelectronic devices. This material is also commercially viable.

Considering that gel electrolytes freeze at low temperatures and dry out at high temperatures, resulting in loss of flexibility and ion transport capacity, Peng et al. [90] prepared an aqueous eutectic hydrogel electrolyte based on a concentrated lithium bis (trifluoromethane) sulfonimide (LiTFSI) solution. As shown in Figure 4a, first the authors synthesized polyacrylamide (PAAm) in LiTFSI solution and then introduced the Fe(CN)_6_^3−/4−^ redox pair after hydrogel formation. Due to the thermocouple effect, the redox ions will react to form a potential difference between the two ends when there is a temperature difference (Figure 4b). Since the LiTFSI ionic compound can regulate the hydration of the hydrogel by changing the concentration of LiTFSI and inhibit the ice crystallization induced by ionic hydration, the freezing resistance and self-humidification ability can be inhibited. This quasi-solid aqueous TEC system containing PAAm/LiTFSI/Fe(CN)_6_^3−/4−^ hydrogel electrolyte was found to work continuously in the temperature range of −15 °C to 70 °C, and the Seebeck coefficient of the system at temperatures of −15 °C and 45 °C was 1.02 mV K^−1^ and 1.29 mV K^−1^, respectively, which is still very high. It also demonstrated long-term environmental stability, with little change after seven days in the environment without encapsulation or packaging.

In parallel with the technological development of thermoelectric materials, the market development of thermoelectric devices also needs to be emphasized. Flexible thermoelectric devices have great application prospects in portable power generation and localized refrigeration. Developed countries such as the United States and Japan have initiated government-funded large-scale thermoelectric research and development projects. The U.S. Department of Energy is working with companies to develop materials for waste heat recovery from high-temperature exhaust gases from automobile engines; South Korea has established a research center for energy recovery systems for advanced hybrid electric vehicles; and China has invested 30 million RMB to develop thermoelectric materials. A study analyzing 2365 TEG patents filed by 28 companies around the world revealed the potential for commercialization of related waste heat recovery products, but based on a preliminary analysis of the International Patent Classification (IPC), existing related patent applications are nearing saturation. It is a challenge to identify niche markets and promote government subsidies to increase the demand for TEG components, thereby reducing the cost of TEG components to expand the market demand for TEG-based WHRS and creating a virtuous cycle for the diffusion of related technologies [91].

## 4. Applications of Photothermoelectric Devices

The investigation into photothermal–thermoelectric devices is of paramount importance for the progression of renewable energy technologies. Such devices possess the capability to harness not only solar thermal energy, via the absorption of sunlight and its subsequent conversion into heat, but also thermoelectric energy, through the transformation of temperature gradients into electricity. This dual functionality presents a unique strategy for enhancing the overall efficacy of solar energy exploitation. The following will categorize the advances in photothermoelectric materials in recent years in terms of applications such as sensors, thermoelectric batteries, generators, wearable fabrics, and multi-effect devices. The performance of photothermoelectric materials in recent years is listed in Table 1.

### 4.1. Sensors

As scientific and technological advancements continue to progress, the demands placed on sensors across various domains have become increasingly stringent. At present, the majority of sensors rely on battery power, necessitating a continuous external power supply. This dependence results in high maintenance costs due to frequent battery replacements and charging, which not only requires substantial labor and resource investment but is also impractical in certain scenarios. Furthermore, the additional circuits required for power sources augment system complexity and increase device size. Consequently, the development of sensors independent of external energy supplies and enhancements in the efficacy of solar energy exploitation are of paramount importance [92,93,94,95].

Bai et al. [96] presented a self-powered, vertically stretchable, and environment-friendly light intensity sensor. This innovative sensor is realized by the integration of an electrolyte-based thermoelectric gel (photothermal layer) with a porous organic carbon sponge (thermoelectric layer), which is composed of a blend of carbon nanotubes and polydimethylsiloxane. The sensor’s upper photothermal layer adeptly absorbs light, transforming it into heat. This process activates the lower thermoelectric layer to produce thermal voltage signals. Owing to the robust correlation between the output voltage and light intensity, the sensor allows for the active monitoring of light intensity through nuanced photothermoelectric conversion. Furthermore, the authors have devised a light intensity monitoring platform that enables user terminals to access real-time light intensity data. It was found that the maximum temperature difference of the photothermal conversion model reached 13.0 K and the open-circuit voltage rose to 21.8 mV after 4 h of exposure to sunlight. Such passive light intensity sensors offer promising applications in areas including artificial breeding, artificial planting, and efforts focusing on energy conservation and emission reduction. Hou et al. [97] developed a hydrogel with robust toughness, high stretchability, and photo-thermo-responsiveness. They prepared an electron- and ion-bound conductive hydrogel by placing highly conductive Mxene nanosheets and Ca^2+^ into the hydrogel, and Mxene and Ca^2+^ endowed the hydrogel with antifreeze properties, respectively. The double-network KMGHCa gel was obtained by Ca^2+^ crosslinked gellan gum (GG-Ca^2+^) forming a double helix structure as the first network and poly (N-hydroxyethyl acrylate) (PHEAA) as the second network, and 3-(trimethoxysilyl) propyl methacrylate (KH570)-modified Mxene (K-Mxene) nanosheets were used as a cross-linking agent and a functional nanofiller. The experimental results showed that the KMGHC gel sensor has a wide sensing range (0–400%), high sensitivity (GF = 4.40), and outstanding photothermal electric properties, which can monitor human movement in real time, power microelectronic devices, build temperature detection systems, and prepare flexible sensors for applications in personalized health monitoring, wearable human–machine interfaces, energy storage, and other fields. 

### 4.2. Thermoelctric Cells

Photothermal devices and thermoelectric cells hold vast prospects in the realm of energy production. Nevertheless, the integration of these two technologies remains a substantial challenge within practical power supplies. Flexible thermoelectric cells possess excellent mechanical adaptability, allowing seamless integration with dynamic interfaces. This quality makes them an ideal choice for wearable power sources [98,99,100,101]. 

Shen et al. [102] introduced the development of a solar-driven photothermoelectric hydrogel featuring an interlocking structure (PTEH-Interlocking), designed to generate stable electricity from solar light as the power source for a mechanical sensor (Figure 5). The thermo-electric hydrogels (TEHs) were synthesized through the crosslinking of polyacrylamide and carboxymethylcellulose, utilizing [Fe(CN)_6_]^3−^/[Fe(CN)_6_]^4−^ as the thermogalvanic redox couple. The PA-PEI-Fe photothermal film was created in situ via crosslinking with pyrogallic acid (PA) and polyethyleneimine (PEI) following the oxidation of [Fe(CN)_6_]^3−^. Notably, pyrogallic acid, which can be derived from the decarboxylation of natural gallic acid, is abundant in phenolic hydroxyl groups, facilitating the formation of the photothermal film. Interestingly, the PA-PEI-Fe photothermal film infiltrated into the TEH, creating a robust interlocking structure at their interface. The dark and dense PA-PEI-Fe photothermal film absorbs sunlight and converts it into heat, simultaneously shielding the redox couple of the thermogalvanic ion pair from UV-induced damage. Crucially, the interlocking structure facilitates the rapid conversion of solar-derived heat into thermoelectric ions, enhancing electricity generation. Consequently, the PTEH-Interlocking exhibits exceptionally stable electricity production and has been successfully employed as a power supply for a mechanical sensor.

In addition to the photothermoelectric effect, there is also a tendency to develop new composites combining the pyroelectric effect. Liu et al. [103] proposed a polar bear-inspired (PBI) self-powered and uncooled broadband photodetector (PD) based on the pyroelectric effect as well as the photothermoelectric effect, which enhances the broadband response by synergizing the two effects. The detectors (PBI_AAO/CdS_asy-Ag PDs) combine a CdS pyroelectric thermoelectric material with a fibrillated zincite structure with an energy-harvesting and heat-storage functional structure (anodic aluminum oxide (AAO)) and are obtained by constructing a pair of asymmetric Ag electrodes (asy-Ag) in conjunction with asymmetric Schottky bonding. The peak current, specific responsivity, and detection rate were all improved by a factor of 11, which was attributed to the synergistic effect of the photothermoelectric and pyroelectric effects. The PDs are capable of detecting thermally radiating objects with temperatures ranging from −10 °C to 80 °C, and more importantly, for low-power long-wave infrared radiation, they still provide excellent response to human radiation when the distance between the human finger and the detector is up to 10 cm. This study provides a novel design for realizing the synergistic effect of the photothermoelectric and pyroelectric effects, which opens up a new avenue for the improvement of the performance of optoelectronic devices and energy harvesters and has a wide range of application potentials for the detection of human radiation and radiation from other objects.

### 4.3. Generators

Thermoelectric generators possess the capability to harness solar thermal energy and transform it into electrical power, thereby exhibiting considerable potential within the realm of photothermal conversion (PTC). Nevertheless, the broad-scale deployment of conventional TEGs in PTC applications is impeded by certain obstacles, which underscore the inherent limitations of their PTC performance. These limitations result in a suboptimal daily thermal-to-electric conversion efficiency and a conspicuous absence of electrical energy storage capacity. As such, there is an imperative to pioneer integrated and synergistic energy technologies tailored to surmount these underlying challenges. The pursuit of high-performance hybrid photothermoelectric generators has become a prominent area of academic inquiry, owing to their capacity to overcome the previously delineated challenges.

Wen et al. [104] designed a flexible hybrid PTEG utilizing a singular mechanism for the concurrent harvesting of radiation and surface thermal energies (Figure 6). The configuration of the hybrid PTEG comprises a flexible substrate, a thermocouple array, and a light-to-thermal conversion layer. While conventional bulk-type materials such as Bi_2_Te_2.7_Se_0.3_ and Bi_2_Te_3_, along with their corresponding alloys, are lauded for their exceptional thermoelectric figure of merit, they inherently lack flexibility. In recent years, organic-based composite materials have been postulated for the fabrication of flexible thermoelectric devices; however, these systems typically exhibit diminished ZT values. To address this shortcoming, the research team employed the thermocouple array by precision printing n-type (Bi_2_Te_2.7_Se_0.3_) and p-type (Sb_2_Te_3_) thermoelectric inks onto a polyimide (PI) flexible substrate, thereby engineering a flexible thermoelectric generator. Moreover, a light-to-thermal conversion layer, composed of a light-absorbing layer in synergy with a light-reflecting layer, was integrated to encapsulate the TEG, thereby functioning as an efficient light radiation energy harvester. Consequently, the research team achieved hybrid energy harvesting of thermal and radiation energies through an elegant three-layer configuration. Organic thermoelectric composites, distinguished by their exceptional processability and heightened thermoelectric conversion efficiency within medium- to low-temperature ranges, have elicited significant interest. As a consequence, the endeavor to develop hybrid organic photothermoelectric generators is deemed imperative. This endeavor not only expands the application horizon of thermoelectric materials but also facilitates the customization of energy conversion efficiency for specific temperature conditions. Tang et al. [23] pioneered ionic liquids to boost PEDOT:PSS conductivity through ion exchange and added surface-charged SiO_2_-NH_2_ (SiO_2_^+^) nanoparticles. These nanoparticles improved IL dispersion in PEDOT:PSS and reduced thermal conductivity, enhancing PEDOT’s conformation and conductive network. As a result, PEDOT:PSS conductivity rose while thermal conductivity fell. The P_IL_SiO_2_^+^ films showed excellent photothermal performance due to their low thermal conductivity, limiting heat loss. Compared to the reference PEDOT:PSS film (~19.7 °C) and P_IL film (~25.31 °C), the P_IL_SiO_2_^+^—5% film achieved a remarkable temperature gain of ~46.15 °C without extra cooling. Introducing 5 wt% SiO_2_^+^ into P_IL increased conductivity to 1172.4 S cm^−1^ and raised the power factor to 33.17 μW m^−1^ K^−2^. This led to a 223% improvement in thermoelectric efficiency over P_IL films. A self-supporting device made from these films generated an output voltage of 0.8 mV and a current of 50 μA under a radiation power of 186.85 mW cm^−2^, with peak power at 13.57 nW, far superior to previous PTE devices. The device also maintained stability, with less than 5% internal resistance change after 2000 bending cycles. This work highlights the potential of PEDOT:PSS films for solar energy conversion and paves the way for future flexible PTE material research.

### 4.4. Wearable Fabric

Flexible thermoelectric devices are well-suited for wearable applications that capture body heat. Organic materials stand out due to their low cost and superior flexibility, making conductive polymers a popular choice. Films based on PEDOT:PSS have attracted attention for their thermoelectric performance but require better mechanical strength. There is a crucial need to develop fibrous generators with both flexibility and robust mechanics. Efforts have been made to improve thermoelectric textiles using precise fabrication techniques, and some focus has been on photothermal textiles, yet investigation into photothermoelectric textiles remains limited.

Zhang et al. [27] introduced a two-step method to create dual-layer photothermoelectric textiles. First, PEDOT:Tos forms the thermoelectric layer, and then a photothermal layer of PPy is added. Customized PPy enhances temperature and photothermoelectric effects through strong near-infrared absorption. A flexible, portable wearable strip was developed using low-conductivity optoelectronic fabric, conductive material, and a textile base. This strip converts body and solar heat, highlighting the potential of textile-based photothermoelectric generators in wearable devices. Zhang et al. [105] further presented a simple, low-temperature polymerization method to create PEDOT-based polyacrylonitrile (PAN) nanofiber films (PAN-PEDOT). This combination optimizes photothermoelectric performance, achieving a power factor of ~256.61 nW m^−1^ K^−2^. A harvester using PAN-PEDOT and copper wires produced 0.17 mV under IR light. Additionally, a flexible solar panel assembled with PAN-PEDOT and conductive nanofibers generated 1.39 mV and 0.56 mV under an IR lamp and sunlight, respectively. This highlights the potential of photothermoelectric fabrics in solar applications.

Photothermoelectric fabrics represent a novel solution capable of continuously converting photothermal energy into thermoelectricity, harvesting wasted body heat, and using solar energy to power devices, thereby reducing energy consumption.

### 4.5. Multi-Effect Devices

Integrating photothermoelectric devices with additional effects, like the hydroelectric phenomenon, paves the way for pioneering applications that are meticulously tailored to a wide spectrum of settings.

Li et al. [106] exploited the combined effects of photothermoelectric and hydroelectric phenomena to engineer a novel wearable photothermoelectric textile capable of harvesting thermal energy from the human microclimate, solar energy, and enthalpy derived from human perspiration. By integrating the elevated Seebeck coefficient and superior electrical conductivity of organic thermoelectric materials with the robust heat absorption attributes of polypyrrole—utilized as an absorption layer for solar thermal energy—the researchers proposed the thermoelectric transduction capabilities of PEDOT in conjunction with inorganic CuI, both functioning as thermoelectric layers with notable thermoelectric efficiency. Polypyrrole, serving as an absorption layer, intensifies thermoelectric conversion by capturing solar thermal energy. The nanoparticulate forms of PPy and CuI, when synergistically interacting with PEDOT, potentially establish an intricate dual-energy filtration system, further enhancing the Seebeck coefficient. As a consequence, the devised photothermoelectric textiles can be donned on the human form, ensuring the generation of elevated voltage outputs in complex environments. 

Multi-effect devices capitalize on renewable energy sources such as human thermal energy, solar energy, and water energy, offering expansive developmental potential and providing viable strategies for the advancement of wearable smart devices.

**Table 1 materials-17-03524-t001:** Recent advances in structural composition materials and properties of photothermal electrical components.

Photothermal Layer	Thermoelectric Layer	Simulated Sunlight Intensity (mW cm^−2^)	Output Voltage and Current
PC sponge/copper foil (CNTs/PDMS)	+TC PVA (Fe^2+^/Fe^3+^/Na/Cl)	100	21.8 mV [96]
Mxene	GG-Ca^2+^/PHEAA/Mxene	2000	—— [97]
PA-PEI-Fe	TEH (PAAm/CMC/Fe(CN)_6_^3−/4−^)	100	>20 mV [102]
PBI_AAO/CdS	Ag electrodes/CdS	Human Finger (a LWIR under low-power radiation)	60.2 pA [103]
Carbon power/PDMS, TiO_2_/PDMS	Bi_2_Te_2.7_Se_0.3_/Sb_2_Te_3_	40	116.2 mV [104]
PEDOT:PSS	P_IL_SiO_2_^+^ (PEDOT: PSS/IL/SiO_2_^+^)	186.8	~0.8 mV, 50 μA [23]
PPy	PEDOT: Tos	Body Heat and Solar	0.667 mV [27]
PAN@SiO_2_ nanofiber film	PEDOT: Tos	Sunlight	0.56 mV [105]
PPy	PEDOT/CuI	Body Heat and Solar	0.51 V [106]

## 5. Summary and Outlook

This paper provides a comprehensive analysis of the recent advancements in the fields of photothermal, thermoelectric, and photothermoelectric effect devices, delving deeply into the underlying principles, applications, and potential market impacts of these technologies. Through a meticulous review and analysis of the current literature, the study offers unique insights into the future trajectories of these domains, including possibilities for technological enhancements, significant challenges faced, and pathways to commercialization. Thermal storage is still a major challenge for photothermal materials, and research is now focusing on how to mix different nanomaterials with the working fluid or change the structure of the materials to improve their photothermal conversion capacity. For thermoelectric and photothermoelectric materials, it is viewed from two aspects. First, for technology, we need low-cost and high-performance materials as well as materials that utilize both photothermal and photovoltaic power generation. Second, for marketization, the main issue is to promote government subsidies and increase market demand, as well as to facilitate research to reduce the cost of thermoelectric materials, thus expanding the use and popularity of thermoelectric materials. Furthermore, the article discusses how interdisciplinary collaboration can propel the development of these sustainable technologies and proposes several practical strategies aimed at fostering the practical implementation of thermophotoelectric technology in wearable smart devices and other contemporary applications.

In recent research, the development of photothermoelectric technology has prompted people to develop stretchable, self-healing, and portable materials that can be used for intelligent mutual sensing devices. In the current environment of the world, the utilization of green and sustainable energy is an unchanging theme, which prompts the combination of future photothermoelectric materials with other energy sources, such as water, wind, biomass, and so on.

In this review, we advocate for the advancement of thermoelectric materials characterized by an elevated Seebeck coefficient, coupled with the progression of photothermal materials that ensure efficient energy conversion. Such advancements can be actualized through the incorporation of avant-garde nanoscale material techniques and sophisticated composite fabrication processes, which are poised to dramatically enhance the thermoelectric efficacy and photothermal conversion rates of these materials. In tandem, we endorse the utilization of pioneering micro and nanofabrication techniques and pioneering design tenets to foster the creation of compact, lightweight photothermal electrical devices. These devices are meticulously engineered for seamless integration with wearable technologies, aiming to fulfill the demands of compatibility and practical application. To fortify the resilience and dependability of these devices in diverse environmental contexts, we underscore the significance of bolstering their endurance against temperature variations, humidity shifts, and mechanical shock. As a capstone, we propose the harnessing of artificial intelligence to enrich human–machine interactivity, aspiring towards the realization of intelligent photothermal electrical devices that boast refined adaptive functionalities and enhanced user engagement experiences.

## Figures and Tables

**Figure 1 materials-17-03524-f001:**
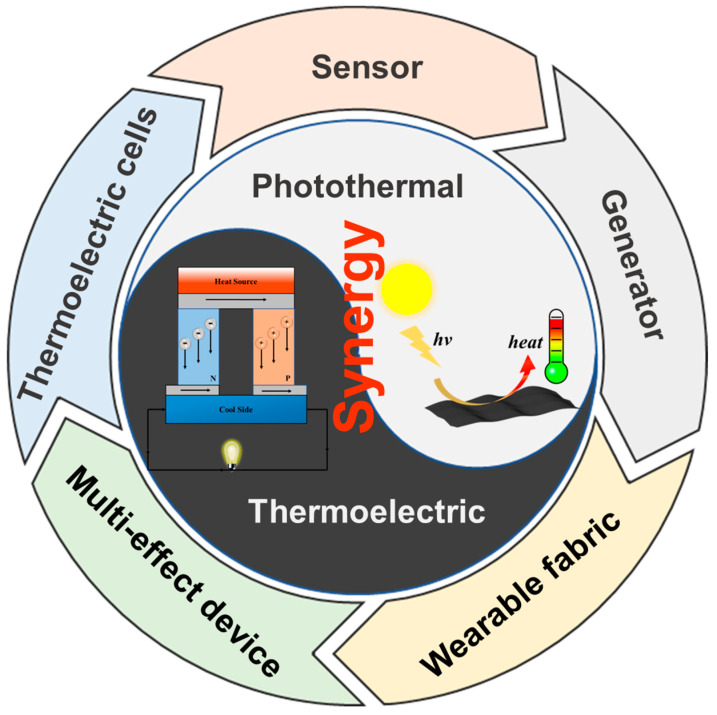
Schematic illustration of materials, mechanisms, and applications of the PTE composites.

**Figure 2 materials-17-03524-f002:**
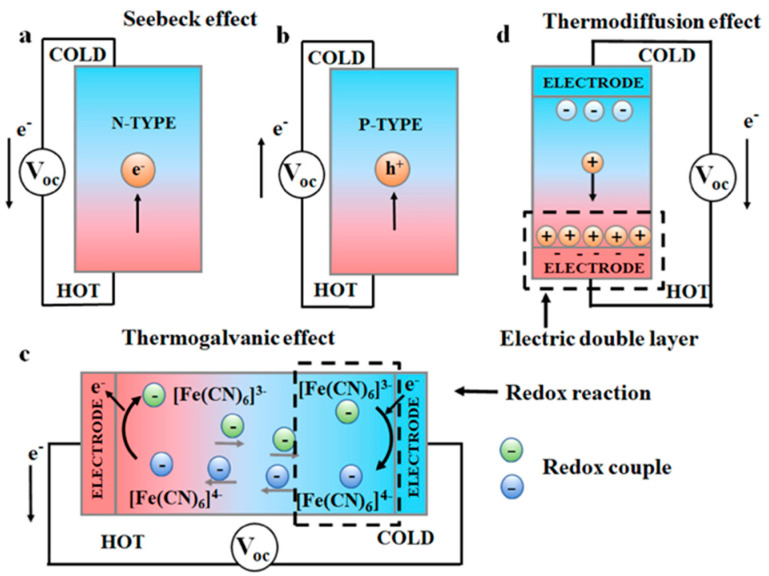
Power generation mechanism of two TE materials. (**a**) n-type and (**b**) p-type e-TE materials based on the Seebeck effect. i-TEs based on (**c**) the thermogalvanic effect and (**d**) the thermodiffusion effect [59] (Copyright from Elsevier).

**Figure 3 materials-17-03524-f003:**
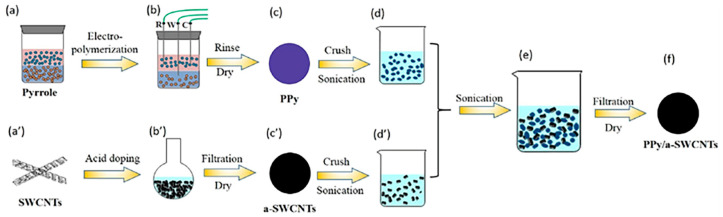
Schematic illustration of the preparation procedure of the PPy/a-SWCNT composites [79] (Copyright from Elsevier).

**Figure 4 materials-17-03524-f004:**
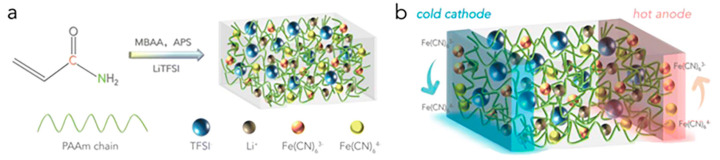
(**a**) Schematic of the formation of the PAAm/LiTFSI-Fe(CN)_6_^3−/4−^ hydrogel electrolyte. (**b**) Schematic showing the working principle of the TEC based on the hydrogel electrolyte [90] (Copyright from the Royal Society of Chemistry).

**Figure 5 materials-17-03524-f005:**
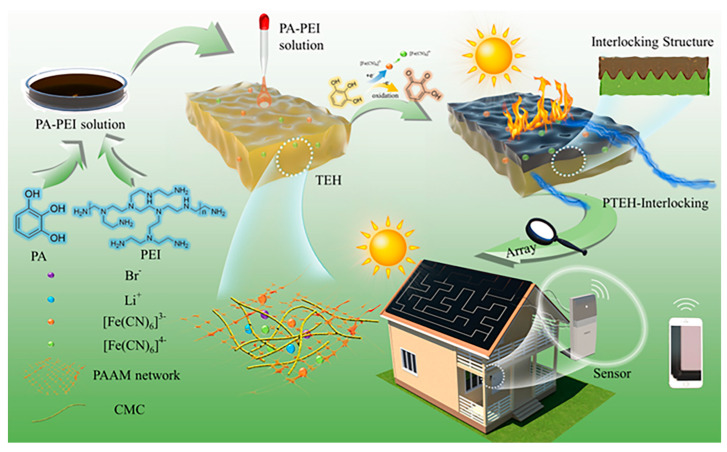
Schematic illustrating the construction of the PTEH-Interlocking and their potential application for photothermoelectric energy conversion [102].

**Figure 6 materials-17-03524-f006:**
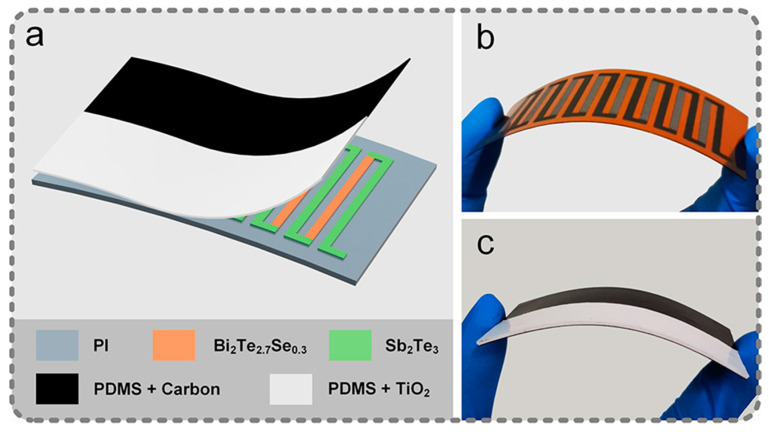
Simple three-layer structure of the developed hybrid photothermoelectric generator (PTEG) illustrated by a schematic diagram and photographic images. (**a**) The developed PTEG is composed of a flexible substrate made of polyimide film (PI), a thermocouple chain made of n-type thermoelectric material (i.e., Bi_2_Te_2.7_Se_0.3_) and p-type thermoelectric material (i.e., Sb_2_Te_3_) to scavenge thermal energy, and a light-to-thermal conversion layer composed of a light-absorbing film and a light-reflecting film to scavenge radiation energy. (**b**) Photograph of the fabricated thermoelectric generator with dimensions of 75 mm × 40 mm × 0.2 mm (eight pairs of thermocouples). (**c**) Photograph of the fabricated PTEG with dimensions of 75 mm × 40 mm × 1 mm after preparing the light-to-thermal conversion layer. These two photographic images demonstrated the good flexibility of the hybrid energy harvester due to its simple three-layer structure and flexible materials [104] (Copyright from the American Chemical Society).

## Data Availability

Not applicable.

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
