# Peer review of "Progress on Material Design and Device Fabrication via Coupling Photothermal Effect with Thermoelectric Effect"

_materials, 2024, doi:10.3390/ma17143524_

Round 1

Reviewer 1 Report

Comments and Suggestions for Authors

1.       The paper's title could be more appealing; it should be revised.

2.       The abstract part needs to be more specific. Please revise the abstract and state clearly how this review is important for society.

3.       The authors should add references for the used equation.

4.       The authors should add and summarize recent developments in a table.

5.       Most of the figures added by authors are not discussed well.

6.       Section 4 needs more relevant further details about the mentioned topics. It’s a review so authors should expand it to provide maximum details to the readers.  

7.       Extensive English corrections are needed.

8.       The summary and outlook section also need to be extended, the authors should add future directions.

Comments on the Quality of English Language

Extensive editing of English language required

Author Response

Dear Reviewer:

Thank you for your valuable and constructive comments concerning our manuscript. We have made corrections based on them and hope the revisions comply with your requirement. Listed below are our responses to your comments.

Reviewer

Comments and Suggestions for Authors

  1. The paper's title could be more appealing; it should be revised.

Reply: Thanks for the advice. The title is changed from “Strategies for Devising Polymer Composites and Applications via Coupling Photothermal Effect with Thermoelectric Effect” to “Progress on Material Design and Device Fabrication via Coupling Photothermal Effect with Thermoelectric Effect”.

  1. The abstract part needs to be more specific. Please revise the abstract and state clearly how this review is important for society.

Reply: In agreement with your proposal, we have enriched the abstract and hope it reflects the importance of the review for the development of society in the summary section.

  1. The authors should add references for the used equation.

Reply: Thanks for your suggestion. We have already included two publications for the equation and added one more for the sake of rigor.

  1. The authors should add and summarize recent developments in a table.

Reply: Thanks to your suggestion. We have added Table 1 listing recent advances in photothermoelectric materials to provide readers with an understanding of the field.

  1. Most of the figures added by authors are not discussed well.

Reply: Thanks for your suggestion and we have discussed in detail Figures 2, 3, 4.

  1. Section 4 needs more relevant further details about the mentioned topics. It’s a review so authors should expand it to provide maximum details to the readers.  

Reply: Thanks for your advice. We have added more relevant advances in this aspect in chapter 4, as well as some additional notes on photo-thermoelectric materials both at the beginning and end of the article, which will hopefully make this article more informative.

  1. Extensive English corrections are needed.

Reply: As indicated in the revised manuscript, the writing has been greatly improved including correction of typos and grammatical errors that appeared in the text.

  1. The summary and outlook section also need to be extended, the authors should add future directions.

Reply: Thank you very much for your suggestion, we have added in the summary and outlook section the future direction of photo-thermoelectric materials and some of the challenges facing now.

Best regards,

Prof. Dr. Cun-Yue Guo

School of Chemical Sciences

University of Chinese Academy of Sciences

Beijing 100049, P. R. China

Tel: +86-10-69672546

Fax: +86-10-69672553

Reviewer 2 Report

Comments and Suggestions for Authors

Liu et al. in their review manuscript titled "Strategies for Devising Polymer Composites and Applications via Coupling Photothermal Effect with Thermoelectric Effect" provided valuable information in  one draft.
It is well organized and written. The article is acceptable with minor revision with following comments.
Please add the information on required ZT, S for the best performance with reference.
Se doped Bi2Te3 act as a n-type material and Sb doped Bi2Te3 also act as a p-type material. So, Please add this information in the text.
Can author add info on the market share of thermoelectric device for sustainable development.

Author Response

Dear Reviewer:

Thank you for your valuable and constructive comments concerning our manuscript. We have made corrections based on them and hope the revisions comply with your requirement. Listed below are our responses to your comments.

Reviewer

Comments and Suggestions for Authors

  1. Please add the information on required ZT, S for the best performance with reference.

Reply:

Sincerely thank you for the advice you have provided, we have added the relevant information on ZT, S, and PF, etc. given in the literature in Chapter 3. We don’t add for the photothermal part and photothermoelectric part as this is not a priority, but there is a new table (Table 1) added for the photothermoelectric part (Chapter 4).

  1. Se doped Bi2Te3 act as a n-type material and Sb doped Bi2Te3 also act as a p-type material. So, Please add this information in the text.

Reply:

Thanks for providing the advice. We have added the relevant information in chapter 3.

  1. Can author add info on the market share of thermoelectric device for sustainable development.

Reply:

Thanks for your constructive comments. We have added the current market development of thermoelectric materials at the end of Chapter 3, which we hope will help the reader to deepen their understanding of the field.

Best regards,

Prof. Dr. Cun-Yue Guo

School of Chemical Sciences

University of Chinese Academy of Sciences

Beijing 100049, P. R. China

Tel: +86-10-69672546

Fax: +86-10-69672553

Reviewer 3 Report

Comments and Suggestions for Authors

The reviewed manuscript is devoted to the describing of polymer-based composites which can be used for converting of solar energy into electrical ine using photothermal and thermoelectric effect. Material presented in the manuscript is interesting and useful for specialists working in this field and manuscript can be published in "Materials", but only after some corrections in it (see below).

1. There a lot of misprints (such as, for example, ".[1,2]" instead "[1,2].") which should be thouroghly corrected.

2. Thermal conductivity in the equation (page 2) and in the text should be designated with one symbol (better as "k" (kappa)).

3. Page 4, lines 125-129, page 5, line 175: It would be better to write about "nanoparticles" instead "nanoparticle".

4. Which abbreviationa should be explained at the first appearance in the text (SDS, i-TE, PSH, etc.).\

5. The first sentence in the part "3.1. Electronic thermoelectric materials" should be rephrased, as Seebeck effect present practically in all the materials (except suprconductors in the superconducting state etc.) not only thermoelectric materials.

6. To the inorganic thermoelectrics belong not only derivatives of bismuth-tellurium selenides-tellurides, but also scutterudites, clathrates, silicides, complex oxides etc. In should be mentioned in the manuscript.

7. References in the text should be given in the one manner, using first (not last) author of corresponding paper.

8. Part "3.3" should be renamed, possibly as "3.3. Thermo-electrochemical cells".

9. Statements given in the sentences presented in the pages 7, lines 255-259, and in the page 8, lines 301-305, should be referenced.

10. The formulae of chemical compounds in the references [9,18,87] should be corrected.

Author Response

Dear Reviewer:

Thank you for your valuable and constructive comments concerning our manuscript. We have made corrections based on them and hope the revisions comply with your requirement. Listed below are our responses to your comments.

Reviewer

Comments and Suggestions for Authors

  1. There a lot of misprints (such as, for example, ".[1,2]" instead "[1,2].") which should be thouroghly corrected.

Reply:

Thank you for your advice, we have corrected the errors that appeared in the text.

  1. Thermal conductivity in the equation (page 2) and in the text should be designated with one symbol (better as "k" (kappa)).

Reply:

Thanks for your patience in correcting my fault, we have changed the upper case Greek letter kappa “Κ” in the formula to lower case ”κ” and the lower case Greek letter kappa is still used in the text.

  1. Page 4, lines 125-129, page 5, line 175: It would be better to write about "nanoparticles" instead "nanoparticle".

Reply:

Thank you very much for your careful review, and we have corrected all the misspellings in these two sentences (along with one we discovered later) to ensure the linguistic accuracy of the article.

  1. Which abbreviationa should be explained at the first appearance in the text (SDS, i-TE, PSH, etc.).\

Reply:

Thanks to your careful review, we have labeled the acronyms SDS, i-TE, PSSH, etc. clearly when it first appeared (Since PSH does not appear in the text, we assume you are trying to refer to PSSH)

  1. The first sentence in the part "3.1. Electronic thermoelectric materials" should be rephrased, as Seebeck effect present practically in all the materials (except suprconductors in the superconducting state etc.) not only thermoelectric materials.

Reply:

Thank you for the correction, we have corrected the expression from “The Seebeck effect is usually present in electronic thermoelectric (e-TE) materials, which are materials that exist under a temperature gradient that results in the generation of a voltage across the material due to the diffusion of electrons and holes” to ”The Seebeck effect is the generation of a voltage in a material due to the diffusion of electrons and holes over a temperature gradient, which is manifested in various types of materials, such as e-TEs.”

  1. To the inorganic thermoelectrics belong not only derivatives of bismuth-tellurium selenides-tellurides, but also scutterudites, clathrates, silicides, complex oxides etc. In should be mentioned in the manuscript.

Reply:

Thanks sincerely for providing the advice, we have added the information in Chapter 3.

  1. References in the text should be given in the one manner, using first (not last) author of corresponding paper.

Reply:

Thanks for pointing that out. We have changed the author all the way to the first one.

  1. Part "3.3" should be renamed, possibly as "3.3. Thermo-electrochemical cells".

Reply:

Thanks for the correction, we also found this problem and corrected it.

  1. Statements given in the sentences presented in the pages 7, lines 255-259, and in the page 8, lines 301-305, should be referenced.

Reply:

Thank you for your suggestions. These two sentences are in the same document since the first sentence or two of their respective paragraphs are in the same document and the cited article has been noted earlier, so it’s unnecessary to quote these two articles again.

  1. The formulae of chemical compounds in the references [9,18,87] should be corrected.

Reply:

Thanks for the advice and we have corrected all the spelling errors of chemical formulas that appeared in the references.

Best regards,

Prof. Dr. Cun-Yue Guo

School of Chemical Sciences

University of Chinese Academy of Sciences

Beijing 100049, P. R. China

Tel: +86-10-69672546

Fax: +86-10-69672553

Round 2

Reviewer 1 Report

Comments and Suggestions for Authors

The revision is Ok. It should be accepted now. 

Comments on the Quality of English Language

Minor editing of English language required